# Artificial Neural Network Modeling for Predicting Wood Moisture Content in High Frequency Vacuum Drying Process

**Haojie Chai [1], Xianming Chen [2], Yingchun Cai [1] and Jingyao Zhao [1,*]**

[1] Ministry of Education, Key Laboratory of Bio-Based Material Science and Technology, College of Material Science and Engineering, Northeast Forestry University, Harbin 150040, China; nefuchj@163.com (H.C.); caiyingchunnefu@163.com (Y.C.)

[2] College of Information and Computer Engineering, Northeast Forestry University, Harbin 150040, China; chenxianming@nefu.edu.cn

[*] Correspondence: zjy_20180328@nefu.edu.cn; Tel./Fax: +86-451-8219-1002

**Abstract:** The moisture content (MC) control is vital in the wood drying process. The study was based on BP (Back Propagation) neural network algorithm to predict the change of wood MC during the drying process of a high frequency vacuum. The data of real-time online measurement were used to construct the model, the drying time, position of measuring point, and internal temperature and pressure of wood as inputs of BP neural network model. The model structure was 4-6-1 and the decision coefficient $R^2$ and Mean squared error (Mse) of the training sample were 0.974 and 0.07355, respectively, indicating that the neural network model had superb generalization ability. Compared with the experimental measurements, the predicted values conformed to the variation law and size of experimental values, and the error was about 2% and the MC prediction error of measurement points along thickness direction was within 2%. Hence, the BP neural network model could successfully simulate and predict the change of wood MC during the high frequency drying process.

**Keywords:** neural network; high frequency drying; moisture content; wood

## 1. Introduction

Wood MC (moisture content) is one of the crucial indicators in the drying process as it has a direct impact on the stability of wood drying quality, and a reasonable control of MC can help in meeting the various quality requirements of actual wood products [1]. High frequency vacuum drying is a joint drying technology with a fast drying rate, low energy consumption, and low environmental pollution [2], and is in widespread use throughout the wood drying industry [3]. However, due to the interference of high frequency electromagnetic fields, the traditional MC online monitoring device cannot be used normally, which makes the online prediction and effective detection of wood MC problematic [4]. Therefore, the research on the prediction model of wood MC is of great significance in the high frequency drying process.

The wood structure is complex and it is difficult to establish a precise mathematical model through mathematical mechanism. An accurate control of MC requires precise mathematical models. The high frequency vacuum drying of wood is a non-linear, complex drying process, which is difficult to accurately express, control, or implement by using general mathematical methods [5]. The concept of BP (Back Propagation) neural network comes from the biological system of brain, which is composed of numerous neurons that are connected to each other through synapses that process information. The neural network has decent characteristics for predicting nonlinear complex systems [6,7], and the model reflects the intrinsic connection of experimental data after a finite

number of iterative calculations. It is not only strong at processing nonlinearity, self-organizing adjustment, adaptive learning, and fault-tolerant anti-noise [8–10] but also can effectively deal with nonlinear and complex fuzzy processes. An effective network prediction model can be established without any assumption or theoretical relationship analysis, based on the historical data and powerful self-organization integration capabilities [11,12].

Artificial neural networks are increasingly being used for modeling in the field of wood science. For instance, in the field of wood drying, Avramidis (2006) [13] predicted the drying rate of wood based on neural network construction model; Zhang Dongyan (2008) [14] constructed a neural network model for predicting wood MC during conventional drying; İlhan Ceylan (2008) [15] used neural network models to study wood drying characteristics; Watanabe (2013, 2014) [16,17] employed artificial neural network model to predict the final moisture content of Sugi (*Cryptomeria japonica*) during drying and evaluate the drying stress on the wood surface. Ozsahin (2017) [18] utilized artificial neural networks to successfully predict the equilibrium moisture content and specific gravity of heat-treated wood. The artificial neural networks are widely used in the study of conventional drying characteristics, stress monitoring, and MC prediction of wood [19]; however, the use of neural networks to predict changes in the wood MC during high frequency drying has been rarely studied.

Hence, in order to provide a predictive model for the control of wood MC during high frequency drying, based on the BP neural network algorithm and using the real-time online measurement data, drying time, location of measuring point, and internal temperature and pressure of wood as the input to neural network model, the changes in the wood MC can be predicted. Also, the feasibility and prediction accuracy of the model was analyzed.

## 2. Materials and Methods

### 2.1. On-Line Monitoring of Wood Internal Temperature and Pressure

Some uniform and defect-free Mongolian pine (*Pinus sylvestris* var. *mongholica* Litv.) were selected. The 200 mm ends were removed at both ends of the test piece, and the specifications were $120 \times 120 \times 500$ mm specimens after sawing and planning, and the initial moisture content was 50%. As shown in Figure 1, five temperature pressure measuring points were uniformly preset at the center of the sample in the thickness direction. Drilling holes on the side of specimen with a 4 mm drill bit to depth of 60 mm (seeing Figure 1 for specific locations). Each measuring point was embedded with one of the pressure and temperature fiber sensors, and the locations where the sensors were in contact with the surface of wood were coated with silica gel to ensure good sealing. The data was recorded online through the optical fiber sensors.

As shown in Figure 2, 1 is drying tank of high frequency vacuum with the diameter of 650 mm and length of 1350 mm; 2 and 4 are upper and lower plates respectively; and 3 is test material. The high frequency generator oscillates at the frequency of 27.12 MHz and outputs the effective power of 1 kW, which is powered by the center of electrode plate length.

During the drying process, the wood control temperature was set to 55 °C, the ambient pressure was set to 8 kPa, and the control of high frequency output time was set to stop for 2 min after a continuous oscillation for 7 min. In the early stage of drying, wood was quickly taken out and weighed after every 4 h, the real-time MC of wood was calculated, and the pressure and temperature values of five measuring points before the sample was taken out were recorded. In the middle stage of drying, the data were recorded once every 8 h; while, in the later stage of drying, the data were recorded once every 12 h.

The drying was carried out for 204 h until wood MC was dried to 11.56%. The experiment was stopped and a total of 135 data were recorded.

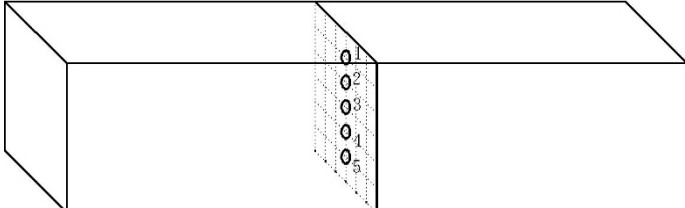

**Figure 1.** Diagram of the wood tested sample and location of the sensors. (1 is the upper layer measuring point; 2 is the upper middle layer measuring point; 3 is the core layer measuring point; 4 is the lower middle layer measuring point; and 5 is the lower layer measuring point).

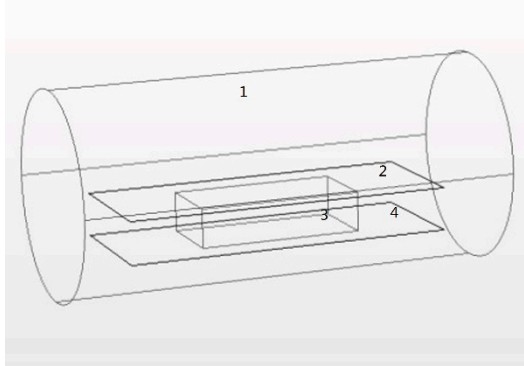

**Figure 2.** Drying tank of high frequency vacuum.

*2.2. BP Neural Network Model*

The BP (Back Propagation) model is currently the most studied and widely used artificial neural network model [20]. It has a powerful nonlinear mapping ability and the qualities of human intelligence such as self-learning, adaptive, associative memory, and parallel information processing. It can imitate the human brain nervous system to store, retrieve, and process the information with an excellent fault tolerance, and is extremely suitable for modeling and control of complex systems [21]. The Python language has a rich and powerful class library. It is an interpreted, interactive, and pure object-oriented scripting programming language that combines the best design principles and ideas of several different languages and is widely used in various fields of software development and application programming. Therefore, this paper built the BP neural network model using Python language programming.

2.2.1. Determination of Neuron Number

The neural network prediction model in this paper uses a three-layer feedforward network structure, which includes an input layer, a hidden layer, and an output layer [22]. The hidden layer can be further divided into a single hidden layer and multiple hidden layer according to the layer number. The multiple hidden layer is composed of multiple single hidden layers. Compared with a single hidden layer, a multiple hidden layer has a stronger generalization ability and higher prediction accuracy, but the training time is longer. The selection of hidden layers should be considered comprehensively based on the network accuracy and training time. For a simple mapping relationship, in case the network accuracy meets the requirements, the single hidden layer can be selected to speed up the process. For a complex mapping relationship, the multiple hidden layer can be selected to enhance the network prediction accuracy. Therefore, according to the research requirements, the study chose a single hidden layer.

The number of hidden neurons also has a certain impact on the network [23]. The neuron number in the hidden layer is directly related to the predictive power of network model. If the number is too high, it will not only increase the network training time but also the network will not converge to the target error, resulting in an over-fitting. If the number is too small, the model training will be

insufficient and would not be able to completely express the relationship between the input variables and output parameters, thus affecting the predictive ability of the model. Therefore, the determination of neuron number in hidden layer is particularly critical [24].

The optimal neurons number was determined via trial and error method [5]. The neuron number in hidden layer was set to 4~10, and the learning error and epoch of different nodes were tested by network training. The optimal node was obtained by comparison analysis.

### 2.2.2. Data Normalization

The data obtained during the experiment were randomly divided into two data sets: a training group and a test group. The 101 test data of the training group accounted for 75% of the total data, while 34 data of the test group accounted for 25% of the total data.

Each input sample usually has different physical meanings and dimensions; hence, in order to make each input sample have an equally important position and also to prevent the adjustment of the weight into the flat area of error, the input sample needs to be normalized [5]. In addition, as the neurons of the BP neural network adopt the Sigmoid transfer function and the output is between [0, 1], it is also necessary to normalize the output samples (Equation (1)).

$$X' = \frac{X - X_{min}}{X_{max} - X_{min}} \tag{1}$$

where $X'$ is the $X$ normalization value; $X_{max}$ and $X_{min}$ are the maximum and minimum values of $X$, respectively.

The neurons in each layer are only connected to the neurons in the adjacent layer and there is no connection between the neurons in each layer. Also, there is no feedback connection between the neurons in each layer. The input signal first propagates forward to the hidden node and then through the transformation function. The output information of the hidden node is propagated to the output node and the output result is given after processing. In general, the Sigmoid transfer function (Equation (2)) is used on all nodes of hidden layer. In the output layer, all nodes use the linear transfer function Pureline.

$$f = \frac{1}{1 + e^{-x}} \tag{2}$$

where $f$ represents the neuron output value and x represents the neuron input value.

### 2.2.3. Model Performance Analysis

In the model correlation test, the model was evaluated by using the determination coefficient $R^2$ and Mse (Mean squared error) of the training sample [25].

The determination coefficient $R^2$ is defined as:

$$R^2 = \frac{\sum_{i=1}^{n} (t_i - p_i)^2}{\sum_{i=1}^{n} t_i^2 \sum_{i=1}^{n} p_i^2}. \tag{3}$$

The Mean square error (Mse) is calculated as [12]:

$$Mse = \frac{1}{n} \sum_{n=1}^{n} (t_i - p_i)^2 \tag{4}$$

where $t_i$ ($i$ = 1, 2, ... , $n$) is the predicted value of the ith sample, $p_i$ ($i$ = 1, 2, ... , $n$) is the true value of the ith sample, and n is the total number of all samples. The decision coefficient is in [0, 1], and the closer the value to 1, the better the model performance, and the closer to 0, the worse the model performance. The smaller the sample Mean square error, the better the prediction performance and the better the model performance. The learning efficiency is set to 0.01.

## 3. Results and Discussion

### 3.1. Determination of Neuron Number

The corresponding relationship between the neuron number of hidden layer and the training error and epoch of neural network is shown in Figure 3. When the node number of hidden layer is 6, the training error is the smallest at 0.07355, and the epoch is 17, the network training is faster. These results show that the neural network model has superb generalization ability at this time [26]; hence, the node number of hidden layer is determined to be 6. According to the node number of hidden layer, the structure of neural network is shown in Figure 4.

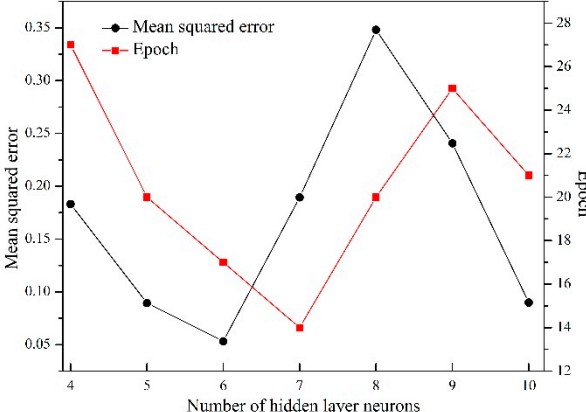

**Figure 3.** Correspondence between the network error and the number of hidden layer neurons.

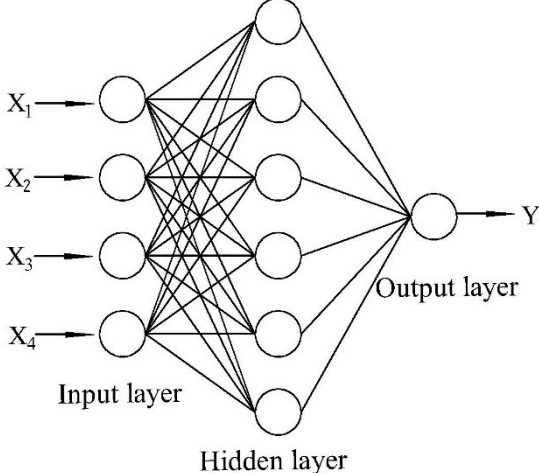

**Figure 4.** BP (Back Propagation) neural network structure diagram ($X_1$: drying time; $X_2$: measuring point position; $X_3$: temperature; $X_4$: pressure; $Y_1$: MC (moisture content)).

### 3.2. Model Performance Analysis

The training regression map for the BP neural network is shown in Figure 5. The linear regression equation between experimental and the predicted value is y = 0.948x + 1.24 while the determination coefficient $R^2$ is 0.974. These results indicate that the experimental and predicted values fit well. The BP neural network model has a good performance and can explain 97% of the above experimental values [24].

The predicted fitted curve for the neural network is shown in Figure 6. The remaining 25% of the samples are predicted and compared with the experimental values. The predicted values are consistent

with the variation and size of the experimental values. Initially, the BP neural network model can simulate and predict the change of wood MC during high frequency drying.

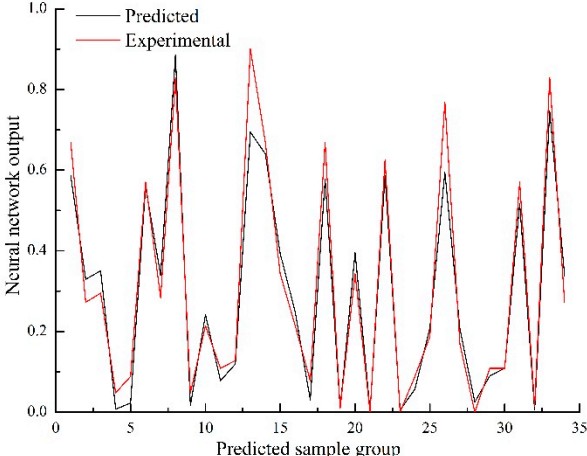

**Figure 5.** Training regression graph of BP neural network.

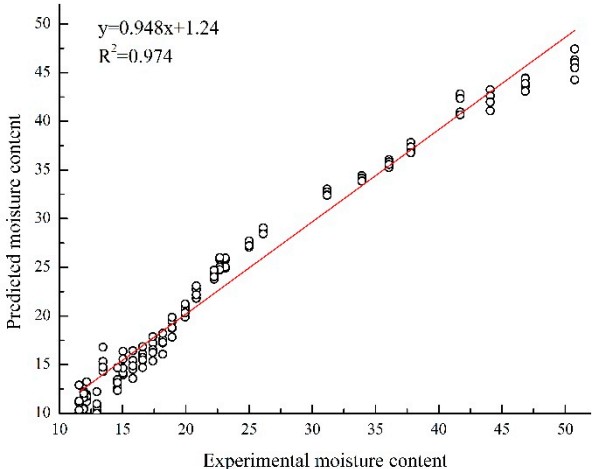

**Figure 6.** Prediction fitting curve of BP neural network.

### 3.3. Prediction of Moisture Content Change

During the drying process of wood, the free water is primarily discharged along the large capillary system above the fiber saturation point. The bound water in cell wall is mainly discharged along the microcapillary system below the fiber saturation point. The bound water is affected by the hydroxyl interaction force in the amorphous region of cell wall [27]. In the early stage of drying, there is a short accelerated drying section. The energy of high frequency radiation is basically used to raise the temperature of wood, and the drying rate is gradually increased from zero. The middle stage of drying is constant-speed drying section. The energy of high frequency radiation is basically used to evaporate the moisture in wood. The MC decreases rapidly and exhibits constant-speed drying tendency. This stage basically completes the evaporation process of moisture in wood. In the later stage of drying, there is less water in wood, and the evaporation rate of moisture and the drying rate of wood gradually decrease [28].

The experimental data is input into the trained model for simulation verification. Figure 7 presents the curve of predicted and experimental values with time. In the early stage of drying, the predicted values are slightly lower than the experimental values; in the middle stage of drying, the predicted values are slightly higher than the experimental values; and in the later stage of drying, the predicted values have a slight wave motion, but overall the value is basically the same as the experimental values.

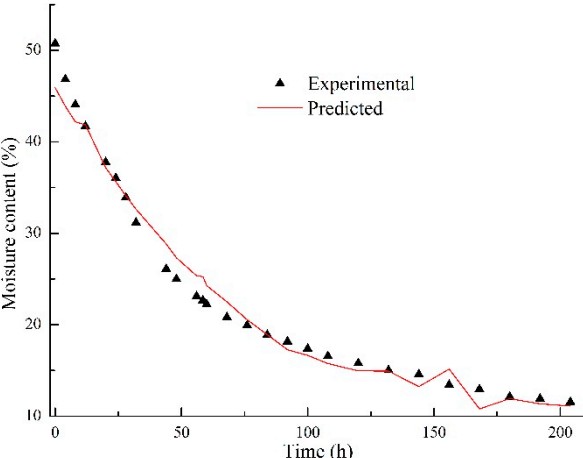

**Figure 7.** Simulation results of BP neural network.

Figure 8 displays the predicted error curve for the neural network (error = experimental value − predicted value [29]). The overall error range is −4%~6% and most of the data is concentrated between −2%~2%, which can basically meet the requirements of prediction accuracy in wood drying.

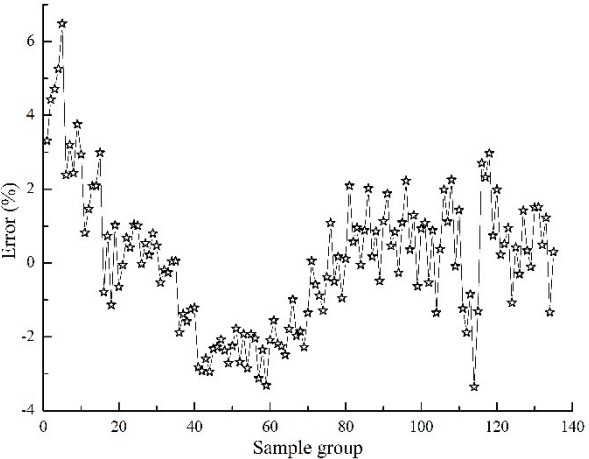

**Figure 8.** Prediction error of BP neural network.

Overall, the predicted data could basically reflect the change trend of MC during the high frequency drying process. The prediction error is about 2%, which proves the feasibility of BP neural network model in MC prediction. Moreover, if the external environmental parameters in the high frequency drying process and the relevant parameters of wood itself are known, the trained neural network model can be used to predict the MC change, thereby eliminating the complicated experimental detection process and saving time and cost [30].

*3.4. Analysis of Stratified Moisture Content Prediction Error*

Figure 9 shows the MC prediction error of measurement points along thickness direction. In the early and later stage of drying, the error is positive and the predicted values are slightly less than the experimental values. In the middle stage of drying, the error is negative and the predicted values are slightly larger than the experimental values. Among these, the error in the middle stage of drying is the largest, followed by the early and later stage of drying. In the early and middle stage of drying, along the thickness direction of test material, from top to bottom, the error increases firstly, then decreases, then increases, and then decreases. The results show M-type trend with no clear law, and the error of the upper surface measurement point is the smallest. In the later stage of drying, along the thickness

direction of the test material, from top to bottom, the error is firstly reduced, then increased, and then decreased, while the error at the upper intermediate layer is the smallest.

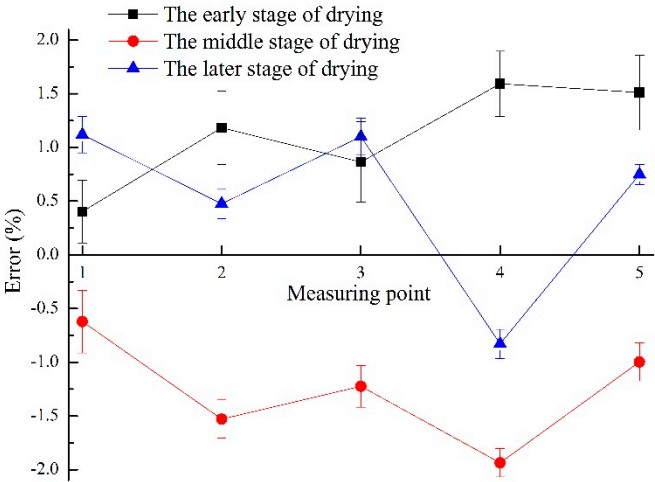

**Figure 9.** Error analysis of stratified moisture content prediction.

Due to the difference in material properties at different locations of the wood and the degree of electromagnetic radiation, the prediction accuracy of each measurement point is different. But overall, the prediction error of MC of each layer is less than 2%, indicating that the prediction accuracy of each measurement point is good and can meet the demand for stratified moisture content prediction.

## 4. Conclusions

The BP neural network was used to simulate the wood MC during the high frequency drying process. The drying time, the location of measuring point, and the internal temperature and pressure of the wood were taken as input variables, while the wood MC was the output variable, 101 test data of the training group accounted for 75% of the total data, while 34 data of the test group accounted for 25% of the total data. The results showed that when the number of hidden layer of neurons was six, the neural network training error was the smallest and the BP neural networks had better stability. The error between the predicted and the experimental values was about 2% and the stratified moisture content prediction error was within 2%, which the model could well simulate the change trend of wood MC during the drying process. In general, although the performance of wood varies greatly and the complex relationship has not been completely elucidated, the proposed neural network model is reliable and has a good predictive power.

**Author Contributions:** H.C. and X.C. conceived and designed the experiments; J.Z. performed the experiments; H.C. and Y.C. analyzed the data and wrote the paper.

**Funding:** The Fundamental Research Funds for the Central Universities (Grant No. 2572018BB08) and the National Natural Science Foundation of China (Grant No. 31670562), financially supported this research.

**Acknowledgments:** The authors thank Zhiqiang Huang for the technical support in the computer field.

**Conflicts of Interest:** The authors declare no conflict of interest.

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
