# Peer review of "Artificial Neural Network Modeling for Predicting Wood Moisture Content in High Frequency Vacuum Drying Process"

_forests, doi:10.3390/f10010016_

Round 1
Reviewer 1 Report
The article as a novelty represents the possibility of using neural networks for determining the wood moisture in the high-frequency vacuum drying of wood. Since the determination of wood moisture content in this drying process is experimentally demanding, this method is an important and original solution. In order to improve the understanding of the article I propose bellow some amendments and changes.
The sentence “The artificial neural networks are widely used in the study in the conventional drying characteristics,…” should be corrected. Please add references! It is expected also to include some of these references in the discussion of this manuscript. It should be mentioned how the used BP neural network model differs from previous models, relating to the structure and reliability.
Ch. 2.1 There is expected some more information regarding the drying experiment – including chamber, characteristics of HF field.
The sentence “Watanabe (2013, 2014) [16,17] employed artificial neural network model to predict the final moisture content of Japanese cedar dryness and evaluate the dry stress on the wood surface.” is not understandable. What is cedar dryness exactly? What is dry stress?
Conclusions of the paper are not properly supported with the experiment and results. Namely, there is no information on number of specimens in the study. The error between predicted and experimental values, which is presented in the conclusion, has to include some statistics (mean?, number of specimens,…)
Several terms in the manuscript are used, which are out of common standard terms of drying technology and wood technology or is not understandable. Please consider the following list:
Drying speed must be changed into drying rate.
dry stress – better to use drying stress
Sorption water is not a proper term. The water, or better water vapor is sorbed, and accumulates in the wood structure. The adsorbed water vapor than stays in the wood structure as bound water in the wood cell walls and as water vapor, which stays in wood cells lumens.
Author Response
Point 1: The sentence “The artificial neural networks are widely used in the study in the conventional drying characteristics,…” should be corrected. Please add references! It is expected also to include some of these references in the discussion of this manuscript. (Page1 Line17-21)
Response 1: Yes, I agree. According to your opinion, add references in this sentence and discussion section. (Page1 Line17-21)
Point 2: It should be mentioned how the used BP neural network model differs from previous models, relating to the structure and reliability. (Page3 Line117)
Response 2: Usually the BP neural network model is built using the MATLAB programming language or toolbox. This paper builds the model based on the Phthon language programming, because the Phthon language combines the best design principles and ideas of many different languages. Detailed description in section 2.2。(Page3 Line117)
Point 3: Ch. 2.1 There is expected some more information regarding the drying experiment – including chamber, characteristics of HF field. (Page2 Line84-87)
Response 3: According to your opinion, I add some information regarding the drying experiment in paper. As shown in Figure 2, 1 is drying tank of high frequency vacuum with the diameter of 650 mm and length of 1350 mm; 2 and 4 are upper and lower plates respectively; and 3 is test material. The high frequency generator oscillates at the frequency of 27.12 MHz and outputs the effective power of 1 kW, which is powered by the center of electrode plate length. (Page2 Line84-87)
Point 4: The sentence “Watanabe (2013, 2014) [16,17] employed artificial neural network model to predict the final moisture content of Japanese cedar dryness and evaluate the dry stress on the wood surface.” is not understandable. What is cedar dryness exactly? What is dry stress? (Page2 Line61)
Response 4: According to your opinion, adjusted this sentence. Watanabe (2013, 2014) [16,17] employed artificial neural network model to predict the final moisture content of Sugi (Cryptomeria japonica) during drying and evaluate the drying stress on the wood surface. The“dry stress”is changed to “drying stress”. (Page2 Line61)
Point 5: Conclusions of the paper are not properly supported with the experiment and results. Namely, there is no information on number of specimens in the study. The error between predicted and experimental values, which is presented in the conclusion, has to include some statistics (mean?, number of specimens,…) (Page8 Line314-326)
Response 5: Yes, I agree. According to your opinion, adjust the content of the conclusions section, and add the information about sample amount. Modify the Figure 9, add the error bar for it. (Page8 Line314-326)
Point 6: Drying speed must be changed into drying rate. dry stress – better to use drying stress.
Response 6: Yes, I agree. According to your opinion, the “drying speed” is changed to “drying rate”, the“dry stress”is changed to “drying stress”.
Point 7: Sorption water is not a proper term. The water, or better water vapor is sorbed, and accumulates in the wood structure. The adsorbed water vapor than stays in the wood structure as bound water in the wood cell walls and as water vapor, which stays in wood cells lumens. (Page6 Line238-239)
Response 7: Yes, I agree. According to your opinion, the “sorption water” is changed to “bound water”. (Page6 Line238-239)
Reviewer 2 Report
Dear Authors,
The material is interesting, congratulations.
In my opinion the information on the wood testing sample preparation and on fixing the sensors is not enough. The information about the wood sample drying process is not sufficient.
The analysis of the results should contain the results of experimental moisture content and pressure measurements and their reference to the ANN simulation. It should not only include the analysis of the consistency of the results.
The Fig. 1 caption is not correct - Diagram of the wood tested sample and location of the sensors
The reference to Fig. 4 (p. 5) is not correct - Fig. 5.

Author Response
Point 1: In my opinion the information on the wood testing sample preparation and on fixing the sensors is not enough. (Page2 Line74-83)
Response 1: Yes, I agree. According to your opinion, add some information about the wood sample preparation and sensor fixing. (Page2 Line74-83)
Point 2: The information about the wood sample drying process is not sufficient. (Page6 Line241-248)
Response 2: Yes, I agree. According to your opinion, add the information for wood sample drying process. In the early stage of drying, there is a short accelerated drying section. The energy of high frequency radiation is basically used to raise the temperature of wood, and the drying rate is gradually increased from zero. The middle stage of drying is constant-speed drying section. The energy of high frequency radiation is basically used to evaporate the moisture in wood. The MC decreases rapidly and exhibits constant-speed drying tendency. This stage basically completes the evaporation process of moisture in wood. In the later stage of drying, there is less water in wood, and the evaporation rate of moisture and the drying rate of wood gradually decrease. (Page6 Line241-248)
Point 3: The analysis of the results should contain the results of experimental moisture content and pressure measurements and their reference to the ANN simulation. It should not only include the analysis of the consistency of the results. (Page7 Line312)
Response 3: The internal pressure detection of wood is of great significance for high frequency vacuum drying. However, the pressure is only an input parameter as ANN in the research, and it is not the main research object, so the pressure change is not analyzed. Based on your comments, some references have been added to the results and discussion sections, and Figure 9 has been modified to add error analysis. (Page7 Line312)
Point 4: The Fig. 1 caption is not correct - Diagram of the wood tested sample and location of the sensors. (Page3 Line103)
Response 4: Yes, I agree. According to your opinion, modify the caption of Figure 1. (Page3 Line103)
Reviewer 3 Report
Dear Authors,
I carefully read your paper “Artificial Neural Network Modeling for Predicting Wood Moisture Content in High Frequency Vacuum Drying Process” and I found it interesting and suitable for being published in Forests.
I have some suggestions and comments.
First of all, an English language revision is necessary because the sentences are often too long and fragmentary or incomplete.
1) In the abstract, for example, the sentence: “In order to develop a prediction model for the MC change during high frequency vacuum drying of wood, based on BP (Back Propagation) neural network algorithm, used the data of real-time online measurement and drying time, the position of the measuring point, and the internal temperature and pressure of wood as inputs of BP neural network model to predict the change of wood MC during the drying process.”, is too long and fragmentary.
This sentence needs revision. It should be we used or …. the data of real-time …. were used.
2) In the Introduction, you write: The wood structure is complex and it difficult to establish a precise mathematical model through a mechanism.
What do you mean? Please clarify
3) Page 2, third line from the bottom: … the data was recorded … It should be … the data were recorded …
4) Results and Discussion. Why you use verbs in the past? I suggest to use verbs in the presents because you are showing and discussing the results.
Author Response
Point 1: In the abstract, for example, the sentence: “In order to develop a prediction model for the MC change during high frequency vacuum drying of wood, based on BP (Back Propagation) neural network algorithm, used the data of real-time online measurement and drying time, the position of the measuring point, and the internal temperature and pressure of wood as inputs of BP neural network model to predict the change of wood MC during the drying process.”, is too long and fragmentary. This sentence needs revision. It should be we used or …. the data of real-time …. were used. (Page1 Line17-21)
Response 1: According to your opinion, I adjust this sentence. The study was based on BP (Back Propagation) neural network algorithm to predict the change of wood MC during drying process of high frequency vacuum. The data of real-time online measurement were used to construct the model, the drying time, position of measuring point, and internal temperature and pressure of wood as inputs of BP neural network model. (Page1 Line17-21)
Point 2: In the Introduction, you write: The wood structure is complex and it difficult to establish a precise mathematical model through a mechanism.
What do you mean? Please clarify. (Page1 Line42)
Response 2: According to your opinion, adjusted this sentence. The wood structure is complex and it is difficult to establish a precise mathematical model through mathematical mechanism. (Page1 Line42)
Point 3: Page 2, third line from the bottom: … the data was recorded … It should be … the data were recorded …(Page3 Line94)
Response 3: According to your opinion, the “was” is changed to “were”. (Page3 Line94)
Point 4: Results and Discussion. Why you use verbs in the past? I suggest to use verbs in the presents because you are showing and discussing the results. (Page8 Line314)
Response 4: Yes, I agree. According to your opinion, I replace verbs in the past with verbs in the presents on Results and Discussion. (Page8 Line314)